# Comparative Transcriptome Analysis Reveals the Effect of *Aurantiochytrium* sp. on Gonadal Development in Zebrafish

**DOI:** 10.3390/ani13152482

**Published:** 2023-08-01

**Authors:** Yanlin Huang, Hao Yang, Yikai Li, Yuwen Guo, Guangli Li, Huapu Chen

**Affiliations:** 1Guangdong Research Center on Reproductive Control and Breeding Technology of Indigenous Valuable Fish Species, Guangdong Provincial Key Laboratory of Pathogenic Biology and Epidemiology for Aquatic Economic Animals, Fisheries College, Guangdong Ocean University, Zhanjiang 524088, China; huangyanlin5534@163.com (Y.H.); lxq02030@gmail.com (H.Y.); gengge5403@163.com (Y.L.); gdougyw@163.com (Y.G.); guangligdou@163.com (G.L.); 2Key Laboratory of Utilization and Conservation for Tropical Marine Bioresources of Ministry of Education, Hainan Key Laboratory for Conservation and Utilization of Tropical Marine Fishery Resources, Yazhou Bay Innovation Institute, Hainan Tropical Ocean University, Sanya 572022, China

**Keywords:** *Aurantiochytrium* sp., gonadal development, zebrafish, transcriptome

## Abstract

**Simple Summary:**

*Aurantiochytrium* sp. has received much attention as a potential resource for large-scale production of omega-3 fatty acids. In this project, we fed zebrafish through feed supplementation for 56 days and found that *Aurantiochytrium* sp. extracts improved the gonadal index of zebrafish. Moreover, it promoted oocyte maturation in an ex vivo environment.

**Abstract:**

*Aurantiochytrium* sp. has received much attention as a potential resource for mass production of omega-3 fatty acids, which contribute to improved growth and reproduction in aquatic animals. In this study, we evaluated the gonadal index changes in zebrafish supplemented with 1–3% *Aurantiochytrium* sp. crude extract (TE) and the effects of ex vivo environmental *Aurantiochytrium* sp. on oocytes. 1% TE group showed significant improvement in the gonadal index, and both in vitro incubation and intraperitoneal injection promoted the maturation of zebrafish oocytes. In contrast, the transcriptome revealed 576 genes that were differentially expressed between the 1% TE group and the control group, including 456 up-regulated genes and 120 down-regulated genes. Kyoto Encyclopedia of Genes and Genomes (KEGG) and Gene Ontology (GO) pathway analysis of differentially expressed genes indicated that *Aurantiochytrium* sp. potentially affects pathways such as lipid metabolism, immune regulation, and oocyte development in zebrafish. The results of this study enriched the knowledge of *Aurantiochytrium* sp. in regulating gonadal development in zebrafish and provided a theoretical basis for its application in aquaculture.

## 1. Introduction

The rise of global trade, dwindling wild fish stocks, competitive product pricing, rising incomes, and urbanization have all contributed to the development of aquaculture [1]. Fishmeal and fish oil were crucial components of aquaculture, as they provide critical mega-3 (n-3) long-chain polyunsaturated fatty acids (LC-PUFA) in feed to support larval growth and gonadal development in subadult fish [2], eicosapentaenoic acid (EPA; 20:5n-3), and docosahexaenoic acid (DHA; 22:6n-3) in particular [3,4]. However, global landings of feed fish were decreasing [1], and the demand for fishmeal and fish oil has been rising [5]. The imbalance between supply and demand has compelled the aquaculture industry to seek alternatives. As the foundation of the marine food cycle, marine microalgae were considered a commercial alternative to high-quality fishmeal and fish oil [6,7].

*Aurantiochytrium* sp. is a heterotrophic unicellular marine eukaryote found in coastal marine habitats across the globe [8]. In coastal regions, they are usually found in decaying algae, mangrove leaf litter, or substrate settings, and their cells amass a high variety of useful bioactive chemicals, such as unsaturated fatty acids, astaxanthin, squalene, etc. [9,10,11]. Unlike fish oil, its production process is not constrained by time or geography, and its production cycle is brief, making it a sustainable resource [7,12,13].

*Aurantiochytrium* sp. has been utilized by corporations in nutraceuticals, infant formula, and dietary supplements [14]. *Aurantiochytrium* sp. biomass products (DHAgold, DSM, Maryland) also were commercially accessible in the aquaculture industry. Numerous academic studies have been undertaken on them. Willer et al. discovered that microencapsulated *Aurantiochytrium* sp. diets improved gonadal development in oysters (*Ostrea edulis*) by containing twelve times more n-3 fatty acids than normal diets [15]. Dietary inclusion of *Aurantiochytrium* sp. at a 2% level considerably increased growth, survival, nutritional quality, and stress tolerance in Penaeus monodon, according to Jaseera et al. [16]. Researchers have shown that *Aurantiochytrium* sp. dietary supplements can reduce like-for-like cannibalism in orange-spotted grouper (*Epinephelus coioides*) by altering 5-HT and cortisol levels in the brain and serum [17]. Few studies have been conducted on the effects of *Aurantiochytrium* sp. on gonadal development [18,19,20,21,22].

Zebrafish (*Danio rerio*) have a wide range of applications in sex determination and differentiation, in vivo and ex vivo stress, pathology, and nutrition, which makes them superior biological models [23,24,25,26,27]. Transcriptomics is a platform for detecting changes in transcripts in defined cells, specific tissues, or species at certain developmental phases, hence revealing molecular changes in biological processes [28,29,30].

As a result, we gave varied quantities of *Aurantiochytrium* sp. extract (TE) to zebrafish during a critical time of gonadal development in order to uncover its mode of action using transcriptomics. We also explored the effect of TE on oocyte maturation by means of in vitro incubation by intraperitoneal injection.

## 2. Materials and Methods

### 2.1. Sample Source and Diet Preparation

Shenzhen University cultivated and extracted *Aurantiochytrium* sp. Szu445 extracts as previously described [31]. The composition of *Aurantiochytrium* extracts, as determined by GC/MS, is presented in Appendix A. Based on prior exploratory tests, the following concentrations of *Aurantiochytrium* sp. Szu445 extract was added to the diet: 0% *w*/*w* TE (Control), 1% *w*/*w* TE (T1), 2% *w*/*w* TE (T2), and 3% *w*/*w* TE (T3). The preparation of feed is discussed in earlier work [32]. The diets contained fifty percent crude protein, ten percent crude fat, eighty-eight percent dry matter, sixteen percent ash, and three percent crude fiber.

The China Zebrafish Resource Center gave AB wild zebrafish that were cultivated in a zebrafish culture system (Shanghai Haisheng Biological Experiment Equipment Co., Ltd., Shanghai, China). During rearing, the photoperiod was maintained at 14:10, the water temperature was maintained at 28 ± 0.5 °C, the pH was maintained between 7.0 and 8.0, the water was continually oxygenated using an oxygenator, and the dissolved oxygen level was maintained above 5.0 mg/L. After parental breeding, one-month-old wild zebrafish (AB) were chosen at random for testing. Two hundred and forty zebrafish were randomly assigned to four groups of three tanks (10 L) with 20 zebrafish per tank. Before feeding began, the length and weight of zebrafish were determined (0.007 ± 0.001 g and 0.90 ± 0.029 cm, respectively).

Zebrafish were fed different diets for 56 days. The daily feed consumption for fish was 5% of their body weight, and they were fed at 9 am and 4 pm. Throughout the eight-week feeding experiment, fish were weighed every two weeks so that the feed weight could be adjusted. Changes in body weight and length of zebrafish in each group during the experiment were shown in Appendix A. The diet was stopped the day before sampling. Randomly selected zebrafish were euthanized with 20 mg/L of eugenol and placed on ice following the experiment. Zebrafish body weight and gonad weight were weighed and recorded, and the gonadal index was calculated (GSI = gonad weight/body weight × 100).

Nine zebrafish gonads were used for RNA extraction, frozen and stored at −80 °C. At the same time, eight zebrafish gonads were fixed for histological observation, half of which were fixed using the Bouin solution and the other half using a 4% paraformaldehyde solution. Animal procedures were performed in strict accordance with the Guide for the Care and Use of Laboratory Animals, and the Animal Research and Ethics Committee of Guangdong Ocean University accepted the protocol (NIH Pub. No. 85-23, revised 1996).

### 2.2. Histological Examination

The gonads were fixed in Bouin solution for 12 h, and the fixative was subsequently eluted with 70% ethanol. Gradient dehydration in a 70% to 100% ethanol solution followed. Finally, tissues were xylene-cleaned and paraffin-embedded. The tissue segment thickness varied between 5–7 μm. Under a light microscope, sections were stained with hematoxylin and eosin (H&E) and examined for histomorphology (Nikon IQ50, Tokyo, Japan).

After 12 h of fixation with paraformaldehyde, the sections were transferred to 30% sucrose for dehydration. The thickness of tissue sections ranged from 8–10 m after being embedded using a frozen sectioning machine (CryoStar NX50 HOVPD, Thermo Fisher Scientific, Waltham, MA, USA). Sections were stained with H&E and examined under a light microscope (Nikon IQ50, Tokyo, Japan) to examine the histomorphology.

### 2.3. The Influence of Varied Aurantiochytrium sp. Concentrations on In Vitro Oocyte Maturation

Depending on the developmental period of the oocytes, full-grown oocytes (0.575–0.625 mm) were selected from the ovaries of female zebrafish under a microscope ruler and pre-cultured in 60% Leibovitz L-15 medium at 26 °C. After two hours, immature oocytes with complete morphological structure were randomly picked. Randomly, 30 immature oocytes were placed in each well of a 12-well plate. Different amounts of *Aurantiochytrium* sp. extracts were added to L-15 medium and 2 mL of configured medium was added to each well of a 12-well plate. Each experimental group was divided into three replicate groups and incubated at 26 °C for 12 h. The effect of the active product on oocyte maturation in vitro was observed by adding 1 g/mL of 17,20-dihydroxy-4-pregnen-3-one (DHP) as a positive control, and follicle rupture was used as an indicator of oocyte maturation when the ooplasm in the follicle changed from opaque to translucent, as observed through a dissecting microscope. Calculating mature cells/fully grown oocytes equals the GVBD index [33].

### 2.4. Intraperitoneal Administration to Assess the Impact of Aurantiochytrium sp. on Oocyte Maturation and Development

A 0.5% *Aurantiochytrium* sp. extract was injected intraperitoneally into sedated adult female fish; saline served as a control. RNA was extracted from gonadal samples collected 3 h, 6 h, and 12 h after injection, and the effect of the active substance on oocyte maturation in zebrafish was determined by real-time polymerase chain reaction analysis of the expression of oocyte growth and maturation factors such as *egf*, *igf3*, and *inhbb*.

### 2.5. RNA-Seq Analysis and Bioinformatics

According to the manufacturer’s recommendations, RNA was extracted from zebrafish gonad tissue using TRIzol (Invitrogen, Carlsbad, CA, USA). Each set contained three replicates, each consisting of three gonads from fish. Nanodrop 2000 was used to evaluate RNA concentration and purity (Thermo Fisher Scientific) RNA integrity was analyzed using the Agilent Bioanalyzer 2100 and the RNA Nano 6000 assay kit (Agilent Technologies, Santa Clara, CA, USA). Bemac Ltd. was responsible for preparing the two cDNA construct libraries and sequencing the transcriptome. Methods for sequencing and analysis were the same as in earlier research [32].

### 2.6. Real-Time Quantitative PCR (RT-qPCR) Validation

TRIzol (Invitrogen, Carlsbad, CA, USA) was used to extract total RNA from gonadal samples, followed by reverse transcription of RNA using the RevertAid First Strand cDNA Synthesis Kit (Thermo Scientific, USA). All primer pairs have a primer5 design (Appendix A). SYBR Premix Ex Taq II (TaKaRa Bio Inc., Shiga, Japan) was utilized in conjunction with the thermocycler methodology outlined below: 95 °C for 30 s, then 40 cycles of 95 °C for 5 s and 60 °C for 30 s. The entire operation was carried out using a LightCycler 480 machine (Roche, Basel, Switzerland). As a housekeeping gene, β-actin served as a reference gene, allowing us to verify standard and normal gene expression. The 2^−ΔΔCt^ technique was utilized to determine relative gene expression levels.

### 2.7. Analytical Statistics

All values are expressed as mean ± standard error (SEM). The data were subjected to one-way ANOVA using the IBM SPSS Statistics 24.0 statistical package (SPSS Inc., Chicago, IL, USA), and the least significant difference (LSD) test was performed to determine the significance of mean differences between groups. At a confidence level of 95% (*p <* 0.05), statistical significance was assessed.

## 3. Results

### 3.1. Effect of Aurantiochytrium sp. Extract on Gonadal Maturation in Zebrafish

Figure 1 depicts the variations in gonadal index for each zebrafish group. The gonadal index was significantly higher in zebrafish exposed to 1% TE (*p <* 0.01). There was no statistically significant difference between the 2% TE group (*p* > 0.05) and the control group. Nevertheless, the gonadal index of the 3% TE group (*p <* 0.05) exhibited a declining tendency. We hypothesize that the inhibitory phenomenon was generated by extracts of *Aurantiochytrium* sp. in high quantities. The experimental results indicate that low concentrations of *Aurantiochytrium* sp. crude extract may enhance the gonadal development of zebrafish, whereas high quantities of the extract will hinder gonadal development. We therefore chose samples from the 1% TE group for further investigation.

To assess the influence of *Aurantiochytrium* sp. extract on the development of the zebrafish ovary, the histology of the female gonads of the control group and the 1% TE feeding group was examined 56 days after feeding. Microscopically examining the amount of oocytes at various developmental stages in the ovary. Figure 2A–C,E–G illustrate the cell types of oocytes at various stages. In the zebrafish ovary, there was no discernible difference between the two types of oocytes. Oil red staining revealed that there was no significant difference between the two in terms of lipid buildup.

### 3.2. Effects of Exposure to Different Concentrations of Active Products for 12 h on In Vitro Oocyte Maturation

The incidence of GVBD in zebrafish oocytes exposed to 1 μg/mL DHP in the positive control group was significantly different from that in the control group (*p <* 0.01), indicating that the oocytes used in the experiment had the ability to be mature. There was no significant difference between the exposed 0.01% TE and 0.1% TE and the control group. Compared with the control group, 0.5% TE significantly increased GVBD (*p <* 0.05), and 1% TE was slightly lower than 0.5% TE, but there was still a big gap compared with DHP.

### 3.3. Intraperitoneal Administration to Evaluate the Effect of Activity on Oocyte Maturation and Development

As can be seen from Figure 3, 0.5% TE significantly promoted GVBD in zebrafish oocytes in vitro. Therefore, we also adopted a concentration of 0.5% injection concentration for intraperitoneal administration. The results are shown in Figure 4. 0.5% TE can up-regulate the expression of *inhbb* within three hours (*p* < 0.05), and then return to normal levels. The gene *egf* was found to be significantly inhibited after 6 h of injection (*p* < 0.05). The results of *igf3* were similar to those of *inhbb*, and both were up-regulated within 3 h (*p <* 0.01). The gene esr1 was not affected by TE injection. The genes *vtg1* and *fgf24* were significantly up-regulated after injection (*p* < 0.05), and the expression of *fgf24* was significantly lower than that of the control group after 12 h (*p* < 0.05).

### 3.4. Transcriptome Analysis

Based on Sequencing By Synthesis (SBS) technology, the RNA of different groups of zebrafish gonadal tissues after 56 days of feeding was sequenced using the Illumina Hiseq high-throughput sequencing platform. After sequencing quality control, a total of 36.63 Gb Clean Data was obtained, and the percentage of Q30 bases in each sample (the percentage of sequences with a sequencing error rate less than 0.1%) was 94.25% or more. The GC content was about 46.87–48.56%. The specific sequencing data statistics are shown in Table 1.

Since the zebrafish genome has been sequenced, its genome is used as a reference for sequence alignment and subsequent analysis (GRCz11_release100). The results of the HISAT2 alignment analysis are shown in Table 2. From the statistics of the alignment results, the alignment efficiency of each sample of Reads to the reference genome ranges from 90.43% to 93.05%. The quality is good and can be used for subsequent analysis of differentially expressed genes. The Reads on the paired pairs were assembled and quantified using StringTie after the paired analysis was completed.

### 3.5. Differential Expression Analysis

The analysis results showed that the expression levels of zebrafish gonadal genes were significantly changed after 1% TE feeding relative to the control group. There were 456 up-regulated genes and 120 down-regulated genes. The expression levels of each gene were clustered according to the difference in expression level (q value) and the fold change in expression level, and a scatter plot was drawn in the form of a volcano plot (Figure 5).

Sixteen DEGs were selected for RT-qPCR validation (Figure 6A), and linear fit analysis was performed with the transcriptome data, and the results are shown in Figure 6B. The trends of expression levels between RNA-Seq and RT-qPCR results were consistent, and the transcriptome data could be used for subsequent analysis. We did hierarchical clustering analysis on the screened differentially expressed genes, and the genes with the same or similar expression patterns were clustered, and the results of differentially expressed gene clustering are shown in Figure 6C, which shows that the reproducibility within groups as well as the overall differences between groups were more significant.

GO functional enrichment analysis was performed for DEGs, which were divided into three main functional categories: cellular component (CC), molecular function (MF), and biological process (BP). Figure 7 shows the enrichment of the top 20 pathways with the lowest *p*-values in each part. extracellular space (GO:0005615), extracellular region (GO:0005576), and hemoglobin complex (GO:0005833) were particularly enriched in CC, while lipid transporter activity (GO:0005319), serine-type endopeptidase activity (GO:0004252), and heme binding (GO:0020037) were enriched in MF. The enrichment coefficients of lipid transporter activity (GO:0005319), serine-type endopeptidase activity (GO:0004252), and heme binding (GO:0020037) in MF ranked the top three, while the GO secondary terms enriched by BP The more significant ones were a cellular response to estrogen stimulus (GO:0071391), negative regulation of endopeptidase activity (GO:0010951), complement activation (GO:0006956), proteolysis (GO:0006508), and lipoprotein metabolic process (GO:0042157). In addition, we noticed that some GO terms related to lipid metabolism were enriched including high-density lipoprotein particle (GO:0034364), chylomicron (GO:0042627), triglyceride lipase activity (GO: 0004806), fatty acid omega-oxidation (GO:0010430), lipid transport (GO:0006869), etc.

The KEGG pathway enrichment analysis of gonadal DEGs showed that 151 pathways were enriched, and the top 20 pathways with the most reliable enrichment significance (i.e., smallest Q value) were Amoebiasis, Complement and coagulation cascades, Prion disease, Systemic lupus erythematosus, Glycolysis/Gluconeogenesis, Fructose and mannose metabolism, Tyrosine metabolism, PPAR signaling pathway, Steroid biosynthesis, and Relaxin signaling pathway, Ferroptosis, Galactose metabolism, Glycine, serine and threonine metabolism, ABC transporters, alpha-Linolenic acid metabolism. The results of up- and down-regulation of the top 20 significant pathways and the number of enriched genes are shown in Figure 8A. The annotation results of differentially expressed genes KEGG were classified according to the type of pathway in KEGG, and the classification graph is shown in Figure 8B. Differentially expressed genes in the gonadal transcriptome of *Aurantiochytrium* sp. fed zebrafish involved cellular processes, environmental information processing, genetic information processing, human diseases, metabolism, and biological systems in six categories, among which the most metabolically enriched pathways include Arachidonic acid metabolism, Steroid biosynthesis and Drug metabolism-cytochrome P450 pathways; under the classification of Organismal Systems, Intestinal immune network for IgA production, Insulin signaling pathway, and Adrenergic signaling in cardiomyocytes were significantly enriched. These functional pathway classifications provide us with insight into the potential pathways of *Aurantiochytrium* sp. extracts on gonadal substance metabolism, immunity, and reproductive function in zebrafish.

## 4. Discussion

### 4.1. Aurantiochytrium sp. Significantly Improves Gonadal Index in Zebrafish

*Aurantiochytrium* sp. contains a variety of bioactive substances, including DHA, EPA, arachidonic acid (ARA), and carotenoids [8]. The GSI of the 1% TE group was significantly increased after 56 days of feeding. There was no significant difference between the 2% TE group and the control group. The 3% TE group was significantly down-regulated, showing a concentration effect of low concentration to promote high-concentration inhibition. We speculate that the significant improvement in gonadal index is related to ARA in *Aurantiochytrium* sp. ARA metabolizes prostaglandins and is involved in steroidogenesis and follicular maturation in fish [34]. In European sea bass (*Perca fluviatilis*), ARA induces 17,20β-dihydroxy-4-pregnen-3-one (DHP) production [35]. In tongue sole (*Cynoglossus semilaevis*), dietary ARA increases testosterone production in mature tongue sole testes [36]. In histological sections, the experimental group and the control group did not show a more mature gonadal period. Perinucleolar oocytes (PO), corticoalveolar oocytes (CAO), early vitellogenic oocytes (EVO), late and mature oocytes (LMO), and post-ovulatory follicles all appeared, and oil red staining did not show differences in lipid accumulation. GSI measures the ratio of gonadal weight to body weight and is often used to compare the reproductive status of individuals or different groups of individuals [37]. In this study, the GSI of zebrafish in the 1% TE group was significantly improved, but there was no significant difference in oocyte maturity between the two groups. They were all mature and reproducible individuals. We believe that the increase in oocytes at each stage led to the difference in GSI, which also means that supplementing 1% TE may bring more mature eggs, which needs to be further verified in subsequent experiments.

### 4.2. Aurantiochytrium sp. Promotes Oocyte Maturation

Germinal vesicle breakdown (GVBD) is the initial step in the final maturation of oocytes into fertilizable eggs, which defines the fecundity of female vertebrates [33]. To investigate the effect of TE on zebrafish oocyte maturation, we selected different concentrations of TE in vitro culture. The results showed that compared with the control group 0.5% TE culture, it showed a higher incidence of GVBD, but lower than DHP-induced GVBD. Therefore, we selected 0.5% TE for intraperitoneal administration and examined genes related to oocyte maturation. Oocyte maturation involves not only pituitary gonadotrophins, but also various local paracrine and autocrine factors, which form a complex communication network within the ovarian follicle, and their regulation is precise and complex [38]. Inhibin beta B chain (*inhbb*) belongs to the TGF-β (transforming growth factor-β) superfamily and is involved in the regulatory function of follicular development. It shows that the ovary increases sharply before ovulation and becomes critical in the late stages of oocyte maturation and ovulation [39,40]. Epidermal growth factor (EGF), an important member of the EGF family [41], is an important paracrine/autocrine factor in the vertebrate ovary and is known to stimulate oocyte maturation [42]. The insulin-like growth factor (IGF) family plays an important role in oocyte maturation, especially gonad-specific IGF-3, which is an important mediator of LH action in zebrafish oocyte maturation. They are dynamically expressed during folliculogenesis, while *igf3* expression reaches its maximum level in follicles at the fully mature stage [43]. Estrogen plays a key role in the maintenance of the ovary in vertebrates, its biological function is exerted through the estrogen receptor (ER), and esr1 is the basis for ovarian maintenance in zebrafish [44]. Oviparous vertebrates produce multiple forms of vitellogenin (Vtg), which is the main source of yolk nutrients [45]. The fibroblast growth factor ligand *fgf24* is required for the proliferation, differentiation, and morphogenesis of early somatic gonads, and his mutant has severely hypoplastic gonads, so it develops into a male [46]. After injection of 0.5% TE for 3 h, the expression levels of *inhbb*, *igf3*, *vtg1*, and *fgf24* in the zebrafish ovary were significantly increased, and *igf3* was extremely significantly improved (*p <* 0.01), which was similar to the gene expression trend at oocyte maturation in previous studies. In this study, *egf* was significantly down-regulated after 6 h. This is similar to the results of Anna Chung-Kwan Tse’s study. The expression of *egf* decreased significantly during the maturation of zebrafish oocytes [47]. The expression of esr1 did not change significantly after injection, indicating that TE may not stimulate oocyte maturation as an estrogen analog. Interestingly, *fgf24* expression decreased after 12 h, which may be regulated by feedback inhibition. Combined with in vitro GVBD, 0.5% TE can indeed promote oocyte maturation through some pathways, and compared with 1% TE in the feeding experiment can improve zebrafish GSI, requiring a lower concentration.

### 4.3. Enriched GO and KEGG Terms

High-throughput RNA sequencing methods (RNA-seq) can generate transcriptome at specific moments, providing insights into gene expression and related molecular pathways under specific conditions [48]. We examined zebrafish gonads fed 1% TE for 56 days. After TE ingestion, zebrafish gonadal DEGs were highly enriched in GO terms extracellular space, extracellular region, cellular response to estrogen stimulus, negative regulation of endopeptidase activity, lipid transporter activity, and serine-type endopeptidase activity. GO terms such as high-density lipoprotein particles, high-density lipoprotein particles, α-amylase activity, oxidoreductase activity, complement binding, oxidation of fatty acid omega, and activation of immune response were significantly upregulated. These terms involve lipid metabolism, starch metabolism, and immune regulation, which suggests that *Aurantiochytrium* sp. not only regulates the lipid metabolism of zebrafish gonads but also participates in the regulation of its immune response. Immunomodulation may be related to n-3 polyunsaturated fatty acids in *Aurantiochytrium* sp. which can act as arachidonic acid antagonists, thereby affecting components of natural and acquired immunity [49].

KEGG pathway analysis showed that DEGs involved six categories: cellular processes, environmental information processing, genetic information processing, human diseases, metabolism, and body systems. Metabolic pathways occupied most of the top 20 pathways with q value as the standard, and their importance could not be ignored. Among them, starch and sucrose metabolism, galactose metabolism, glycine, serine, threonine metabolism, and α-linolenic acid metabolism were significantly up-regulated. Glycolysis/glucose production, fructose and mannose metabolism, tyrosine metabolism, steroid biosynthesis, carbon metabolism, and caffeine metabolism were both up-regulated and down-regulated. They are involved in carbohydrate metabolism, lipid metabolism, and amino acid metabolism, and contain the metabolism of three major nutrients. Similar to the GO annotation, KEGG is also enriched in immune-related pathways: complement and coagulation cascades, T cell receptor signaling pathways, and NOD-like receptor signaling pathways. Not only that, signaling pathways such as PPAR signaling pathway, FoxO signaling pathway, and insulin signaling pathway are also described. This indicates that the impact of *Aurantiochytrium* sp. on zebrafish is multi-dimensional.

### 4.4. Transcriptome Reveals Lipid Metabolism Effects

Lipid metabolism is crucial for every stage of gonadal development, especially lipid deposition is necessary for oocyte maturation in teleost fish [36,50]. Apolipoproteins are the major components of lipoproteins and play a physiological role in lipoprotein metabolism. Apoc2, an essential activator of lipoprotein lipase, plays a role in yolk utilization during early zebrafish development and is a component of high-density lipoproteins that may lead to changes in lipoprotein classes during vitellogenesis in fish [51,52]. Carboxyl ester lipase (Cel), also known as bile salt-dependent lipase or bile salt-stimulated lipase, is a lipolytic enzyme secreted by the pancreas [53]. Yaqiqiqiu et al. found that the knockdown of *cel.1* and *cel.2* in zebrafish leads to developmental delay and yolk sac retention, and *cel.1* and *cel.2* may be involved in yolk lipid utilization during zebrafish embryogenesis [54]. Transcriptome results showed that the expression levels of *apoc2*, *apoeb*, *cel.1*, and *cel.2* were up-regulated to varying degrees, indicating that *Aurantiochytrium* sp. may regulate oocyte vitellogenesis and improve yolk lipid utilization during embryogenesis, which is extremely beneficial for offspring development. Teleost fishes can synthesize a major omega-3 long-chain polyunsaturated fatty acid, DHA, from dietary α-linolenic acid through elongation enzymes of very-long-chain fatty acids (Elovl) and fatty acid desaturases (Fads) [55]. DHA is selectively retained during embryogenesis, indicating the importance of this fatty acid for developing embryos and larvae [56]. Fatty acid desaturase 2 (Fads2) belongs to the fatty acid desaturase protein family and catalyzes the first desaturation reaction in LC-PUFA synthesis [57]. Fads2 acts as a Δ6 desaturase on C24 PUFAs, enabling them to synthesize DHA via the Sprecher pathway [58]. Elovl2 is the main elongation enzyme for the endogenous synthesis of DHA from its precursor EPA [55]. Complement to *Aurantiochytrium* sp. Later, the expression levels of *fads2* and *elovl2* were significantly upregulated, reflecting the improved ability of zebrafish to synthesize DHA. As a representative member of n-3 LC-PUFAs, DHA is indispensable for the normal development of the early retina and nervous system of juvenile fish [59].

### 4.5. Transcriptome Reveals Oocyte Development Effects

ADCY6 encodes a protein that belongs to the adenylyl cyclase family and is responsible for the synthesis of cAMP [60]. During oocyte growth, high cAMP concentrations are maintained intracellularly, preventing oocyte prematurity. Oviparous vertebrates produce multiple forms of vitellogenin (Vtg), which is the main source of yolk nutrients [61]. Vtgs synthesized in the ovary may be more efficiently transported to nearby vitellogenin oocytes than those secreted from the liver. Therefore, ovarian-derived Vtgs can also be used as an important complementary source of Vtgs for rapidly growing oocytes. In this study, *adcy6*, *vtg1*, *vtg2*, *vtg3*, and *vtg7* were all significantly upregulated, suggesting that *Aurantiochytrium* sp. may promote vitellogenin accumulation by inhibiting oocyte prematurity, resulting in better-quality mature eggs.

## 5. Conclusions

In this study, we found that feeding *Aurantiochytrium* sp. can improve zebrafish gonadal index and promote oocyte maturation in vitro and in vivo. Transcriptome sequencing explored the role of *Aurantiochytrium* sp. in regulating gonadal development in zebrafish. Differential expression analysis uncovered 576 DEGs. Enrichment pathway analysis revealed that DEGs were significantly enriched in lipid metabolism, immune regulation, and oocyte development. The results of this study provide knowledge of *Aurantiochytrium* sp. in regulating gonadal development in zebrafish, providing a theoretical basis for its application in aquaculture.

## Figures and Tables

**Figure 1 animals-13-02482-f001:**
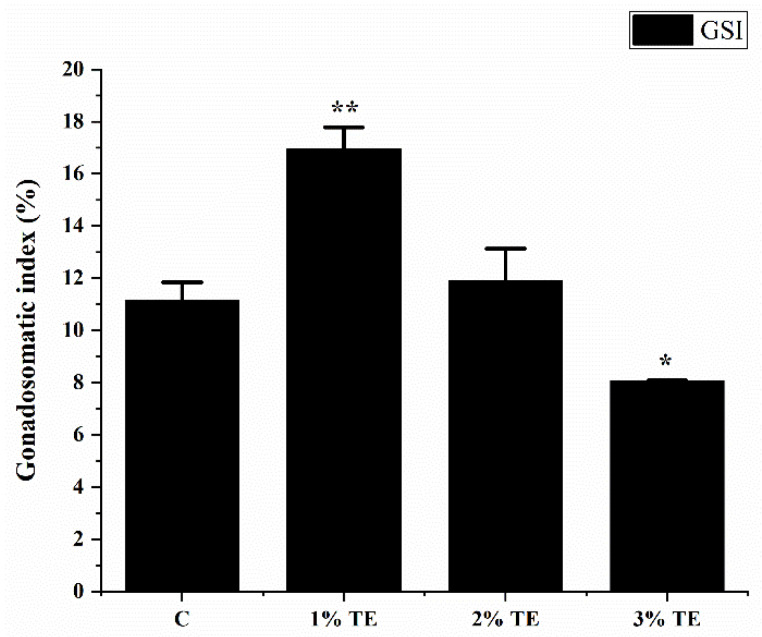
Changes in zebrafish gonadal index (GSI) after feeding *Aurantiochytrium* sp. for 56 days. Each bar represents the mean ± SEM (*n =* 3). Statistically significant differences are expressed as * (* *p <* 0.05, ** *p <* 0.01).

**Figure 2 animals-13-02482-f002:**
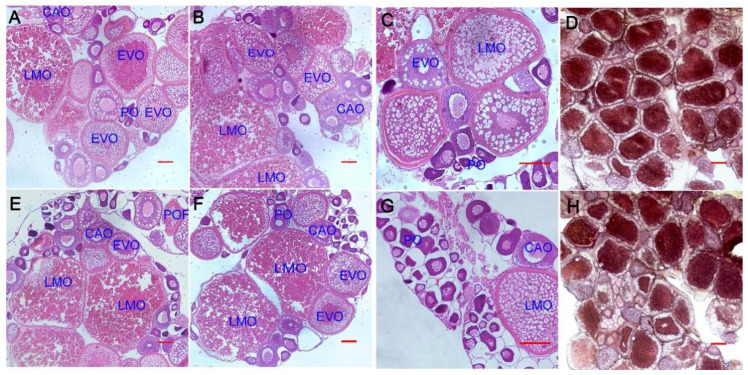
Histological analysis of zebrafish fed 1% *Aurantiochytrium* sp. Extract for 56 days. (**A**–**D**) is the 1% TE group, (**E**–**H**) is the control group, where (**D**) and (**H**) are oil red stains, and the rest are H&E stains. Blue letters indicate oocyte development stages, including perinucleolar oocytes (PO), corticoalveolar oocytes (CAO), early vitellogenic oocytes (EVO), late and mature oocytes (LMO), and post-ovulatory follicles (POF). Scale: 100 μm.

**Figure 3 animals-13-02482-f003:**
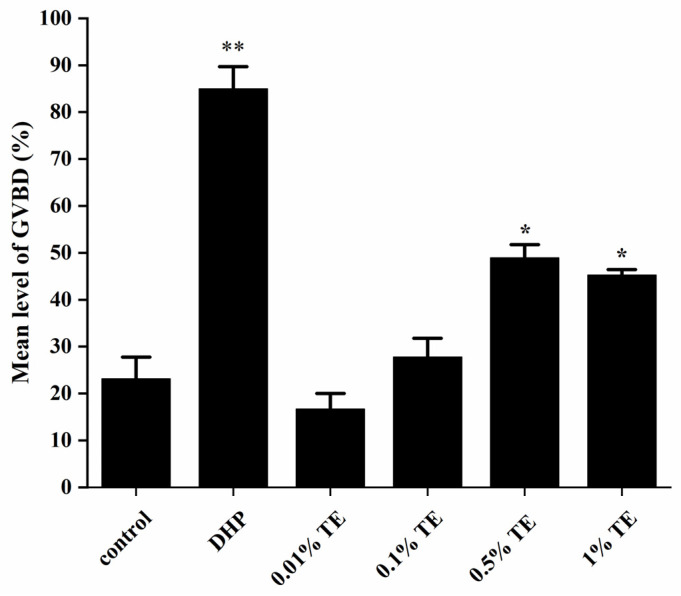
Germinal vesicle breakdown (GVBD) in zebrafish oocytes exposed to different concentrations of *Aurantiochytrium* sp. active product (TE) and 17α, 20β-dihydroxy-4-progesterone-3-one (DHP). Each bar represents the mean ± SEM (*n =* 3). The statistically significant difference is expressed as * (* *p <* 0.05, ** *p <* 0.01).

**Figure 4 animals-13-02482-f004:**
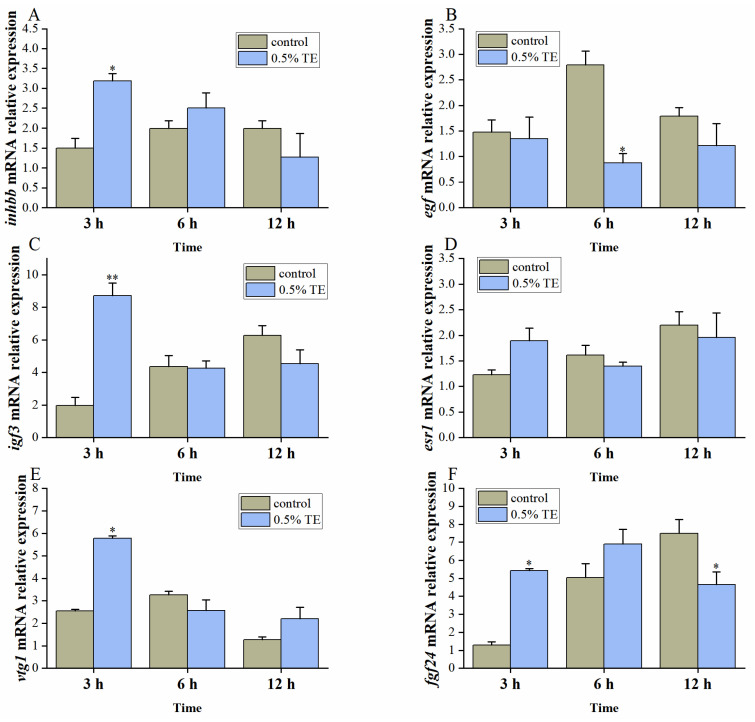
The effect of 0.5% TE on the expression of genes related to oocyte maturation in zebrafish in vivo (**A**–**F**). The data are expressed as mean ± standard error (SEM) (*n =* 3). The statistically significant difference is expressed as * (* *p <* 0.05, ** *p <* 0.01).

**Figure 5 animals-13-02482-f005:**
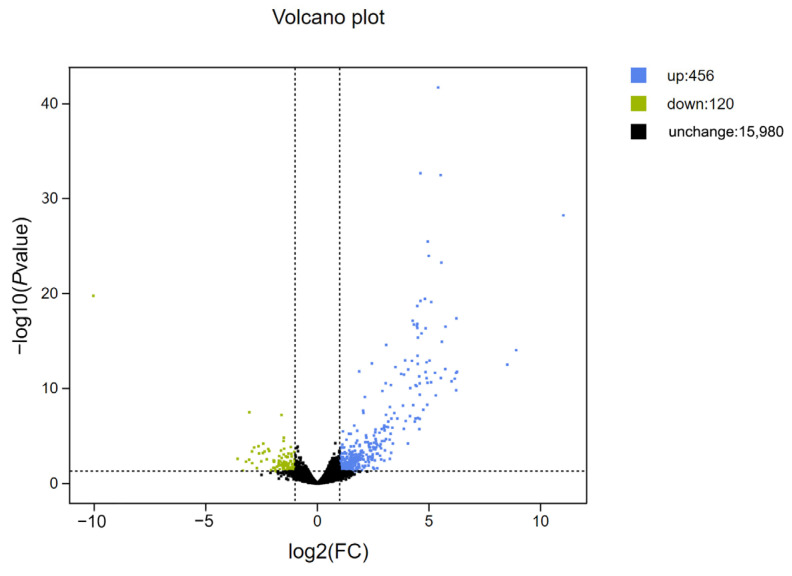
Differential expression of gonadal genes in the control group and 1% TE group. The green dots represent down-regulated differentially expressed genes, the blue dots represent up-regulated differentially expressed genes, and the black dots represent non-differentially expressed genes. The significance criteria are q < 0.05 and Fold Change ≥ 2.

**Figure 6 animals-13-02482-f006:**
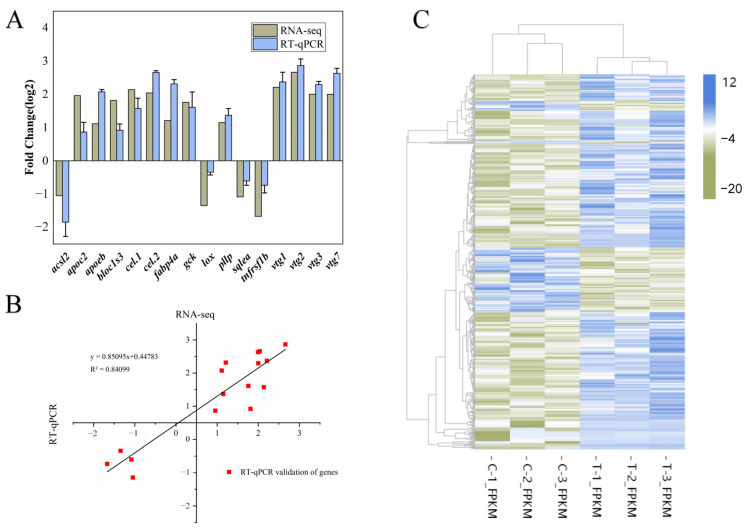
Transcriptome differential gene validation and its cluster analysis (**A**) RT-qPCR differential expression verification. The error line indicates SEM (*n =* 3). (**B**) The consistency of log2 folds changes between RNA-Seq data (x-axis) and RT-qPCR analysis (y-axis). (**C**) Differentially expressed gene expression pattern cluster diagram, used to illustrate the overall pattern of gene expression in different gonadal samples.

**Figure 7 animals-13-02482-f007:**
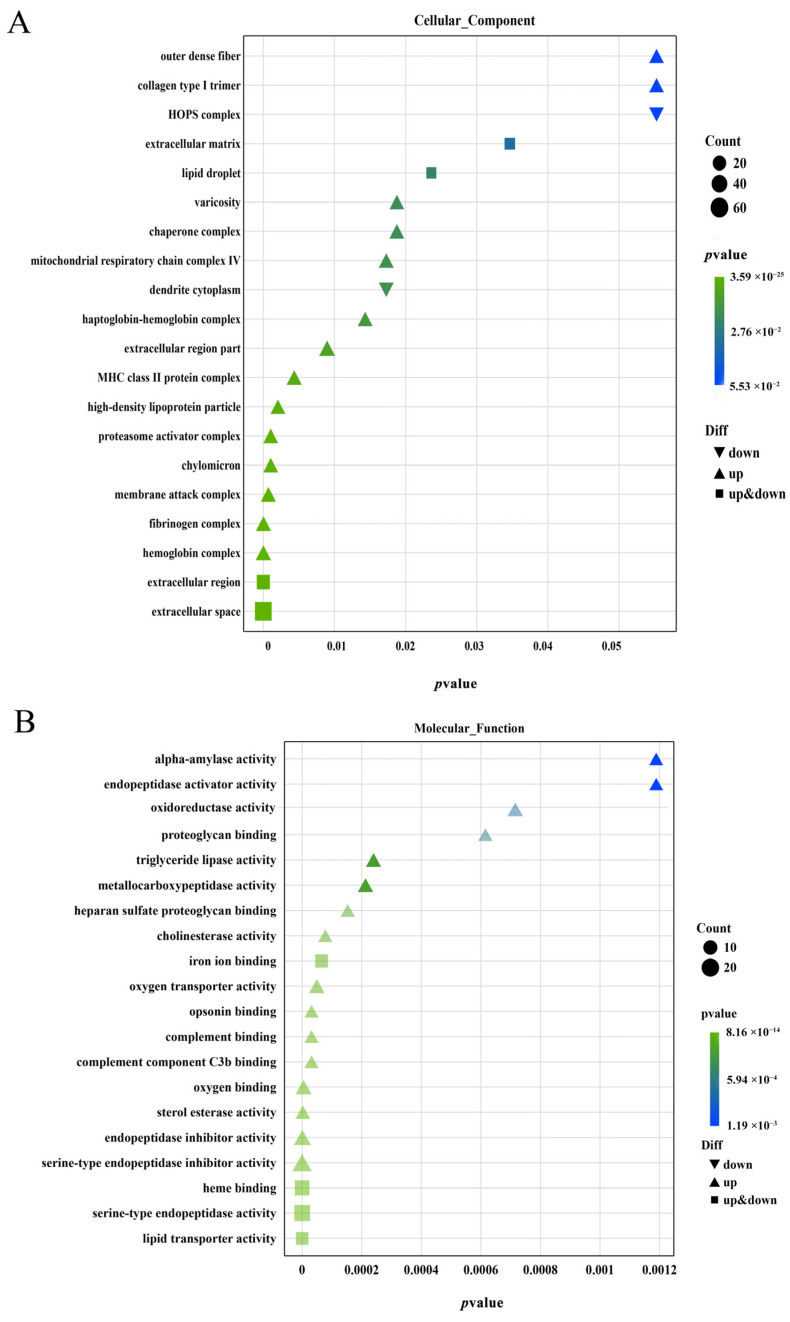
Functional classification of gonadal differentially expressed genes GO Enrichment bubble map (**A**); Cell component category (**B**); Molecular function category; and (**C**) Biological process category.

**Figure 8 animals-13-02482-f008:**
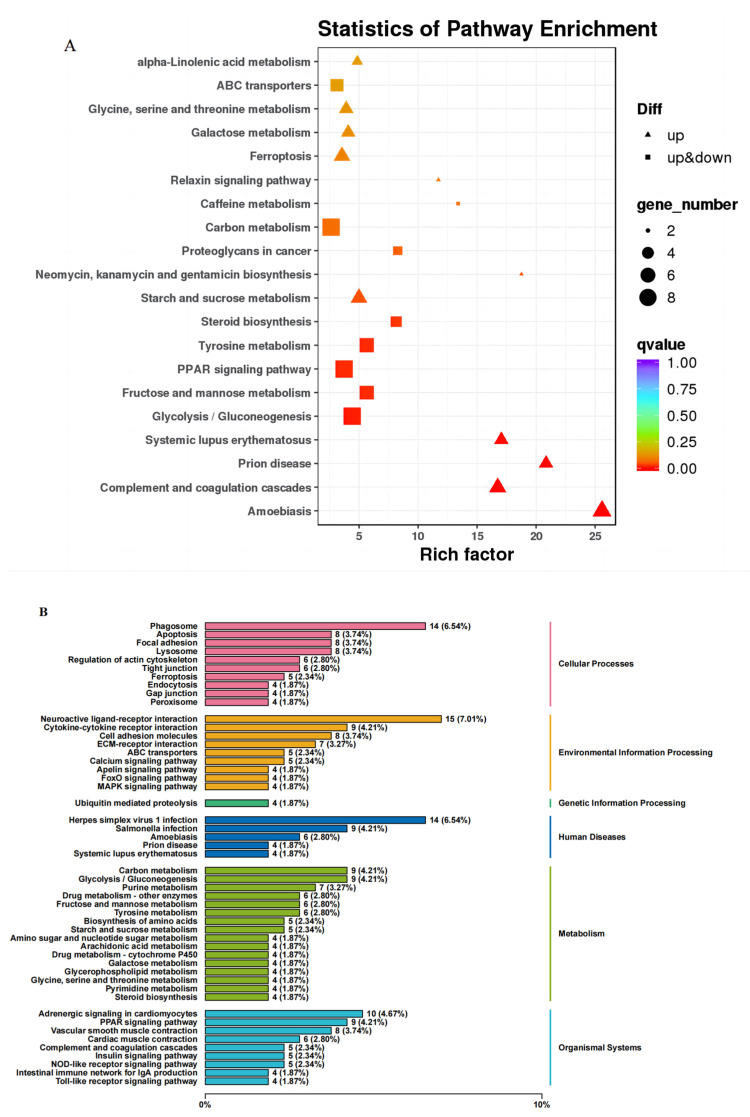
(**A**) Scatter plot of KEGG pathway enrichment of differentially expressed genes, the vertical coordinate indicates the pathway name, the horizontal coordinate is the Enrichment Factor, the color represents the q-value; q-value is the *p*-value after correction for multiple hypothesis testing, and the size of the circle indicates the number of enriched genes in the pathway. (**B**) Differentially expressed genes KEGG classification diagram, the vertical axis is the name of the KEGG metabolic pathway, the left is the pathway name, and the right is the corresponding classification category of each pathway. The same column color indicates the same category.

**Table 1 animals-13-02482-t001:** Sequencing data statistics table.

Samples	Clean Reads	Clean Bases	GC Content	% ≥ Q30
C-1	19,651,945	5,868,622,898	48.21%	94.49%
C-2	20,082,524	5,996,732,510	47.29%	94.89%
C-3	20,863,105	6,234,771,674	47.43%	94.40%
T-1	21,676,987	6,470,986,448	48.56%	94.72%
T-2	20,607,217	6,158,537,846	46.87%	94.25%
T-3	19,747,954	5,902,895,264	46.98%	95.09%

**Table 2 animals-13-02482-t002:** Statistics of sequence comparison of sample sequencing data with the selected reference genome.

Sample	Total Reads	Mapped Reads	Uniq Mapped Reads	Multiple Map Reads	Reads Map to ‘+’	Reads Map to ‘−’
C-1	39,303,890	36,104,705 (91.86%)	33,739,060 (85.84%)	2,365,645 (6.02%)	19,980,679 (50.84%)	19,979,652 (50.83%)
C-2	40,165,048	36,322,349 (90.43%)	34,429,375 (85.72%)	1,892,974 (4.71%)	19,675,070 (48.99%)	19,676,792 (48.99%)
C-3	41,726,210	38,826,693 (93.05%)	30,444,653 (72.96%)	8,382,040 (20.09%)	27,825,562 (66.69%)	27,711,937 (66.41%)
T-1	43,353,974	39,867,193 (91.96%)	32,660,972 (75.34%)	7,206,221 (16.62%)	27,219,252 (62.78%)	27,168,567 (62.67%)
T-2	41,214,434	37,442,521 (90.85%)	34,316,396 (83.26%)	3,126,125 (7.59%)	21,448,304 (52.04%)	21,472,425 (52.10%)
T-3	39,495,908	36,737,497 (93.02%)	32,701,282 (82.80%)	4,036,215 (10.22%)	22,016,744 (55.74%)	21,991,167 (55.68%)

## Data Availability

The data presented in this study are available on request from the corresponding author. The data are not publicly available due to the agreement with funding bodies.

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
