# Peer review of "Comparative Transcriptome Analysis Reveals the Effect of *Aurantiochytrium* sp. on Gonadal Development in Zebrafish"

_animals, 2023, doi:10.3390/ani13152482_

Round 1

Reviewer 1 Report

The authors evaluated the effects of feeding Aurantiochytrium sp. on zebrafish gonad development through examining the gonadal index, oocyte maturation in vitro and in vivo, related RNA expressions and so on. This is a piece of interesting work. However, there are many points needed to be verified or improved in this manuscript before it is acceptable as a publication in the Journal.

Major points:

1. The authors examined the gonadal index after 56 days of feeding Aurantiochytrium sp. That is to say, Aurantiochytrium sp can affect the gonadal index after a long term feeding, then, the authors should explain why the oocytes maturation and genes expressions were examined use the samples just after a acute treatment of TE, but not use the samples after 56 days treatment?

  Maybe, more significant changes induced by TE were discovered in the samples after 56 days treatment.

2. As for the examination of developing oocytes, the authors should directly counting the intact oocytes of different size but not just observe the limited sections.

Minor points

The authors should read through the manuscript and improve the writing since there are many typosinappropriate word and grammar erros.

Examples:

1. In line 23, whats the meaning gonadal changes?  It should be gonadal index.

2. In line 70-71, the sentence is largely unclear.

3. In line 107, whats the meaning of ‘Different quantities of L-15 medium...

4. In the section 2.4, Typos: ...between 5-7 m...’;‘tissue sections ranged from 8-10 m.

Quality of English Language is needed to be improved.

Reviewer 2 Report

The authors have evaluated the Aurantiochytrium sp. on gonadal development of Zebra fish. They have considered different exposure roots (dietary, in vitro, in vivo etc.) and analyse the results well. They have measured the effect of extract on GSI, GVBD, some marker genes, found out the DEGs with interesting bioinformatic analyses. They have written the discussion and conclusion very interestingly. 

But I have some issues / suggestions mentioned in the attached file. The authors are requested to change / correct those points accordingly. I also suggest to change the title little bit for suitable explanation of their work. 

Major revision suggested. 

Need minor modifications/ editing.

Round 2

Reviewer 1 Report

This manuscript is acceptable now after revision.